DOI: 10.1038/s41467-018-03783-y | OPEN

# Quantitative characterization of all single amino acid variants of a viral capsid-based drug delivery vehicle

Emily C. Hartman[1], Christopher M. Jakobson[2,5], Andrew H. Favor[1], Marco J. Lobba[1], Ester Álvarez-Benedicto[1], Matthew B. Francis[1,3] & Danielle Tullman-Ercek[4]

Self-assembling proteins are critical to biological systems and industrial technologies, but predicting how mutations affect self-assembly remains a significant challenge. Here, we report a technique, termed SyMAPS (Systematic Mutation and Assembled Particle Selection), that can be used to characterize the assembly competency of all single amino acid variants of a self-assembling viral structural protein. SyMAPS studies on the MS2 bacteriophage coat protein revealed a high-resolution fitness landscape that challenges some conventional assumptions of protein engineering. An additional round of selection identified a previously unknown variant (CP[T71H]) that is stable at neutral pH but less tolerant to acidic conditions than the wild-type coat protein. The capsids formed by this variant could be more amenable to disassembly in late endosomes or early lysosomes—a feature that is advantageous for delivery applications. In addition to providing a mutability blueprint for virus-like particles, SyMAPS can be readily applied to other self-assembling proteins.

[1] Department of Chemistry, University of California, Berkeley, CA 94720-1460, USA. [2] Department of Chemical and Biomolecular Engineering, University of California, Berkeley, CA 94720-1460, USA. [3] Materials Sciences Division, Lawrence Berkeley National Laboratories, Berkeley, CA 94720-1460, USA. [4] Department of Chemical and Biological Engineering, Northwestern University, 2145 Sheridan Road, Technological Institute E136, Evanston, IL 60208-3120, USA. [5] Present address: Department of Chemical and Systems Biology, Stanford University School of Medicine, Stanford, CA 94305, USA. Correspondence and requests for materials should be addressed to M.B.F. (email: mbfrancis@berkeley.edu) or to D.T-E. (email: ercek@northwestern.edu)

Protein self-assembly relies on optimally balanced energetics that arise in part from the complex interplay of amino acids in apposed protein monomers[1]. Remarkable progress has been achieved using computational methods to understand these interactions, yielding several compelling examples of designed closed-shell structures[2–5] and encapsulated enzymes[6,7]. However, the subtly cooperative nature of these interactions still makes it difficult to predict how particular amino acid substitutions will affect self-assembly behavior. Furthermore, single amino acid substitutions can lead to significant changes to the structure and function of a protein or protein assembly, often leading to surprising outcomes, further complicating computational predictions[8–10]. As such, engineering particles with specifically desired assembly properties remains a challenging goal.

Protein fitness landscapes can provide a useful complementary tool by describing the ways in which systematic changes in primary sequence alter the resulting self-assembly competency[11–13]. In a protein fitness landscape, all possible variants of a protein sequences are ordered such that primary sequences differ only by single amino acid mutations, and variant effect on a functional output is quantified[12]. To date, most quantified fitness landscapes have been determined for enzymes or proteins with a straightforward selection or screen, where fitness is defined as catalytic activity[14,15], binding[16,17], growth[18], or fluorescence[19]. Fitness landscapes have also been explored for a variety of viruses, including human pathogens like HIV, influenza, polio, and others, using infectivity as the selection criterion[20–25]. Recently, a synthetic icosahedral protein assembly was engineered to encapsulate its own genome, and its fitness landscape was evaluated[26]. Several amino acid substitutions yielded improved genome packaging, serum stability, and circulation behavior compared to the original synthetic nucleocapsid. Viral fitness landscapes can also be used to predict vaccine candidates and characterize viral evolution[27], though the infectivity selection strategy combines protein structure, replication, and attachment into a single selection step. Viral fitness landscapes can be generated using bioinformatics techniques, but this requires vast sequence information[21], ruling out its use for little-sequenced viruses, such as bacteriophages or zoonotic pathogens.

Here, we describe a library generation and single-step selection strategy—termed SyMAPS (Systematic Mutation and Assembled Particle Selection)—to study the structure a self-assembling protein capsid composed of a noninfectious viral structural protein, or virus-like particle (VLP). This selection does not rely on infectivity, clinical abundance, or serum stability, and therefore enables experimental characterization of all single amino acid variants of MS2 bacteriophage coat protein (MS2 CP). The resulting fitness landscape is a fundamental roadmap to altering the MS2 CP to achieve tunable chemical and physical properties. After recapitulating the results of many previous investigations in a single experiment[28–30], we separately calculate the effect of ten physical properties on the apparent fitness landscape (AFL), and we evaluate the validity of several common protein engineering assumptions. An additional round of selection identifies a previously unknown variant, CP [T71H], that exhibits acid-sensitive properties that are promising for engineering controlled endosomal release of cargo in targeted drug delivery. The library of MS2 variants can be subjected to future selections to address any number of additional engineering goals. In addition, SyMAPS is a straightforward approach that can be applied more broadly to assess the fitness landscapes of the coat proteins of clinically relevant pathogens, including hepatitis B and human papillomavirus virions[31].

## Results

**Generating a virus-like particle fitness landscape.** MS2 bacteriophage is a well-studied single-stranded RNA virus[32]. MS2 VLPs are composed of 180 copies of a single coat protein, which adopts three conformations (A, B, and C) to form a quasi-equivalent $T = 3$ protein shell. The assembly, structure, and utility of this particle have been well characterized[28,29,33–37], making it an ideal candidate to map its fitness landscape. In addition, MS2 VLPs are an attractive target as they are promising vehicles to deliver small molecule, nucleic acid, or protein cargo to host cells[28,35,36,38]. As MS2 VLPs are used primarily as scaffolding in targeted drug delivery, selections based on infectivity are less relevant in this context. In addition, infectivity selections can fail to identify variants that are non-infectious but have useful physical properties. For instance, we recently discovered with a MS2 CP variant that confers a smaller capsid geometry but is incapable of encapsulating its native genome[10]. As scaffolds, MS2 VLPs are biocompatible, homogeneous, and stable to high temperatures ($Tm = 68\,^{\circ}\text{C}$) and a wide pH range (pH 3–10)[39–41]. The interior cavity is available to load cargo, protecting it from the external environment, while the exterior can be chemically modified to display targeting groups, such as peptides or antibodies[29,42]. Studies have shown that these VLPs are surprisingly long-lived in the bloodstream of mice, stable to serum, and accumulate in several tissues of interest[42,43]. When overexpressed in a bacterial host, the MS2 CP can spontaneously self-assemble into VLPs.

While the MS2 CP is best-known for a sequence-specific, high-affinity protein—RNA interaction between the CP interior and a short RNA stem loop known as the translational operator (TR-RNA)[44], these particles can also nucleate nonspecifically on available negatively charged material to form a VLP. This has been demonstrated for DNA[45], negatively charged proteins[38,46], anionic polymers[38], and anionic nanoparticles[47]. To measure the fitness landscape of the MS2 CP, we harnessed this behavior to establish a genotype-to-phenotype link. This results in the encapsulation of a sample of the mRNA strands inside the cell, including the strands that encode the coat protein itself, if capsid assembly can occur. We then selected for well-formed particles using a size-based purification to separate assembled particles from dimers, truncations, aggregates, and unencapsidated nucleic acids. This strategy was validated using a non-assembling CP variant, a truncated variant, and wild-type MS2 (Supplementary Fig. 1).

To generate a SyMAPS fitness landscape, we first synthesized and characterized a targeted library of all single amino acid variant of the MS2 CP. Every codon in the *MS2 cp* gene was separately swapped for a degenerate NNK codon, which encodes for all twenty amino acids and one stop codon, using a modified version of EMPIRIC cloning[48]. By coupling comprehensive codon mutagenesis with high-throughput sequencing, we quantified MS2 CP variant abundance before and after our size-based selection (Fig. 1). The targeted library contained 2580 possible variants. Of these, over 90% were identified in our plasmid library prior to applying selective pressure. Following selective pressure, 15% of the library was not identified during sequencing—meaning, these variants were selected against and thus are not expected to express and/or assemble into well-formed particles. Around 75% were identified in our assembled virus-like particle library, though relative abundances changed between these two conditions.

Under this selective pressure, we expected assembly-competent variants to increase in relative abundance after a size selection, while non-assembling variants were expected to decrease in abundance. By comparing the percent abundance of every variant across three biological replicates, we generated a quantitative AFL, which contains an apparent fitness score (AFS) for every variant at every residue across the backbone of the self-assembling MS2 CP (Fig. 2, Supplementary Data 1). Positive

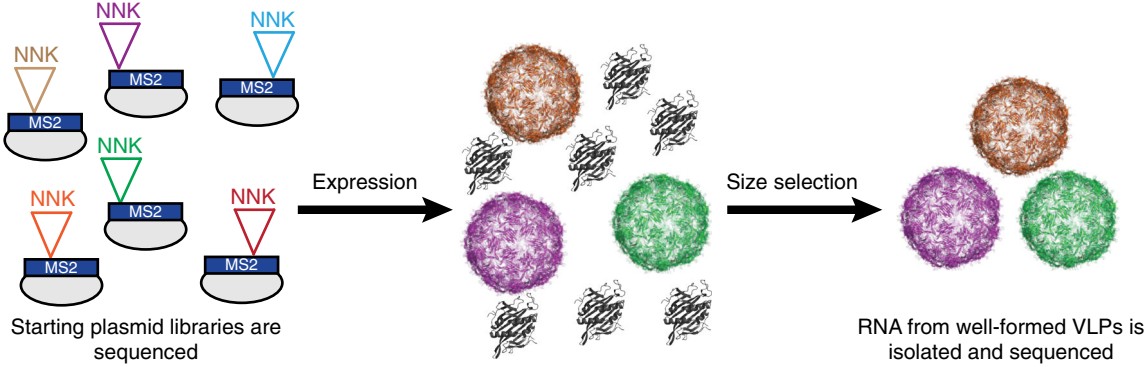

**Fig. 1** The SyMAPS approach to understanding VLP self-assembly. The NNK codon = (Any)(Any)(G/T), and results in the systematic incorporation of all 20 amino acids and a stop codon in each position of the sequence. FPLC SEC was used as a size selection, enriching for well-formed VLPs

AFS (blue) indicate increased variant abundance following selection, while negative scores (red) indicate decreased abundance. Standard deviations were calculated across the three biological replicates (Supplementary Fig. 2).

**Fitness landscape reflects biophysical expectations**. To validate our fitness landscape, we interpreted silent and nonsense mutations as positive and negative controls, respectively. Silent mutations encode for wild-type amino acids and should exhibit a wild type-like phenotype, resulting in a positive AFS. In contrast, nonsense mutations insert stop codons in the coding region of the MS2 CP, and these mutations should produce truncations and result in a negative AFS. We found that the average AFS of silent mutations in our fitness landscape was 0.63, with a standard deviation of 0.3. The average AFS of nonsense mutations was $-2.66$ with a standard deviation of 1.6 (Fig. 3a), significantly lower than the mean AFS of silent mutations ($p < 10^{-4}$ by Student's two-tailed $T$-test).

A specific region of the MS2 CP with secondary structure that should produce characteristic mutability patterns was also validated. The MS2 CP is rich in beta sheets, and beta sheet G is positioned such that one face of the sheet is oriented toward solvent, while the other is oriented toward the protein core (Fig. 3c). Odd-numbered positions, which face solvent, were expected to accommodate charged residues much more easily than the even-numbered, core-facing residues. The AFL shows that lysine and arginine are indeed only tolerated at water-facing positions (Fig. 3b). In contrast, leucine is tolerated at every position along this beta sheet, while glycine and proline, which are both expected to disrupt beta sheets, are poorly tolerated at 5 of 6 and 6 of 6 positions in this region, respectively. These results match biological intuition and validate the selection strategy.

**Mutability index identifies highly mutable residues**. To quantify the mutability of each residue, the Shannon entropy[49]—a measure of diversity at a given residue—was calculated as a proxy for mutability. Comparing Shannon entropy before and after our selection yielded the Mutability index (MI) of each residue (Fig. 4a-b, Supplementary Data 2). This mutability score enables the identification of attractive sites to rationally engineer properties of interest, since positions with higher MI tolerate a wider array of amino acids at that position.

Across the backbone of the MS2 CP, we see a range of MI values, though the average Shannon entropy before selection is higher than the average post-selection Shannon entropy, as expected (Supplementary Data 3). On the exterior of the VLP, several residues are identified as highly mutable, including residue 19. Previous studies successfully installed unnatural amino acids at position 19, validating this finding[29]. On the VLP interior,

several positions are highly mutable (Fig. 4c). One of these residues is position 87, where a reactive cysteine is commonly installed to load small molecule cargo, such as drugs or imaging agents[50]. Perhaps the most mutable region of the MS2 CP contains residues facing the pore, in what is known as the FG loop. In some ways, this is surprising: like other quasi-equivalent structures, the MS2 CP must engage in a conformational change to form three monomeric structures known as the A, B, and C monomers in order to form a closed icosahedral structure[51]. During this process, the FG loop engages in a critical conformational shift, including a *cis/trans* isomerization at Pro-78[52]. The loop is highly flexible, and this flexibility is critical for VLP assembly[34]. Previous work in our lab successfully mutated this region to alter the kinetics of small molecule transport, and others also have successfully manipulated this region[53,54]. Indeed, this mutability indicates that region may be a useful site to insert small, flexible peptides into the MS2 CP.

Our experimental results can be used to understand the parameters that are critical for self-assembly of biomacromolecules like VLPs. To this end, the MI of each residue was compared to its accessible surface area (ASA), which measures the exposure of the residue to solvent. ASA has been used previously to identify mutable, exterior-facing residues[55]. Interestingly, ASA did not globally correlate with MI (Supplementary Fig. 3a-b). However, trends in a region of secondary structure did match between these two metrics. In beta sheet G, which shows a clear difference in mutability between solvent-facing and core-facing residues, similar patterns were observed between MI and ASA, as calculated by PDBePISA (Supplementary Fig. 3c)[56]. Although accessibility does contribute to mutability, additional factors contribute to the complex quaternary structure of this VLP. This result underscores the importance of experimentally generating protein fitness landscapes, particularly for self-assembling structural proteins like VLPs.

**Apparent fitness score confirms VLP assembly**. To be a valuable engineering tool, the AFL should enable a priori insight of whether a MS2 CP variant will self-assemble into a well-formed VLP, even if the mutation is not at an intuitive residue. To confirm this, sixteen example variants with an AFS from $-1.6$ to 0.8 were selected, with an emphasis on proline mutations, which are often difficult to insert into a complex structure, and cysteine mutations, which are useful for bioconjugation. These variants were evaluated for VLP formation. Three of the four proline mutations were expected to form VLPs and one was not. CP[V67P] has an AFS of 0.6 despite inserting a proline into the middle of a beta sheet—a change canonically expected to be highly disruptive to protein structure. CP [E76P] was permitted but CP[L77P] was not, and both are close to the conserved Pro78, which adopts a *cis* conformation in the B

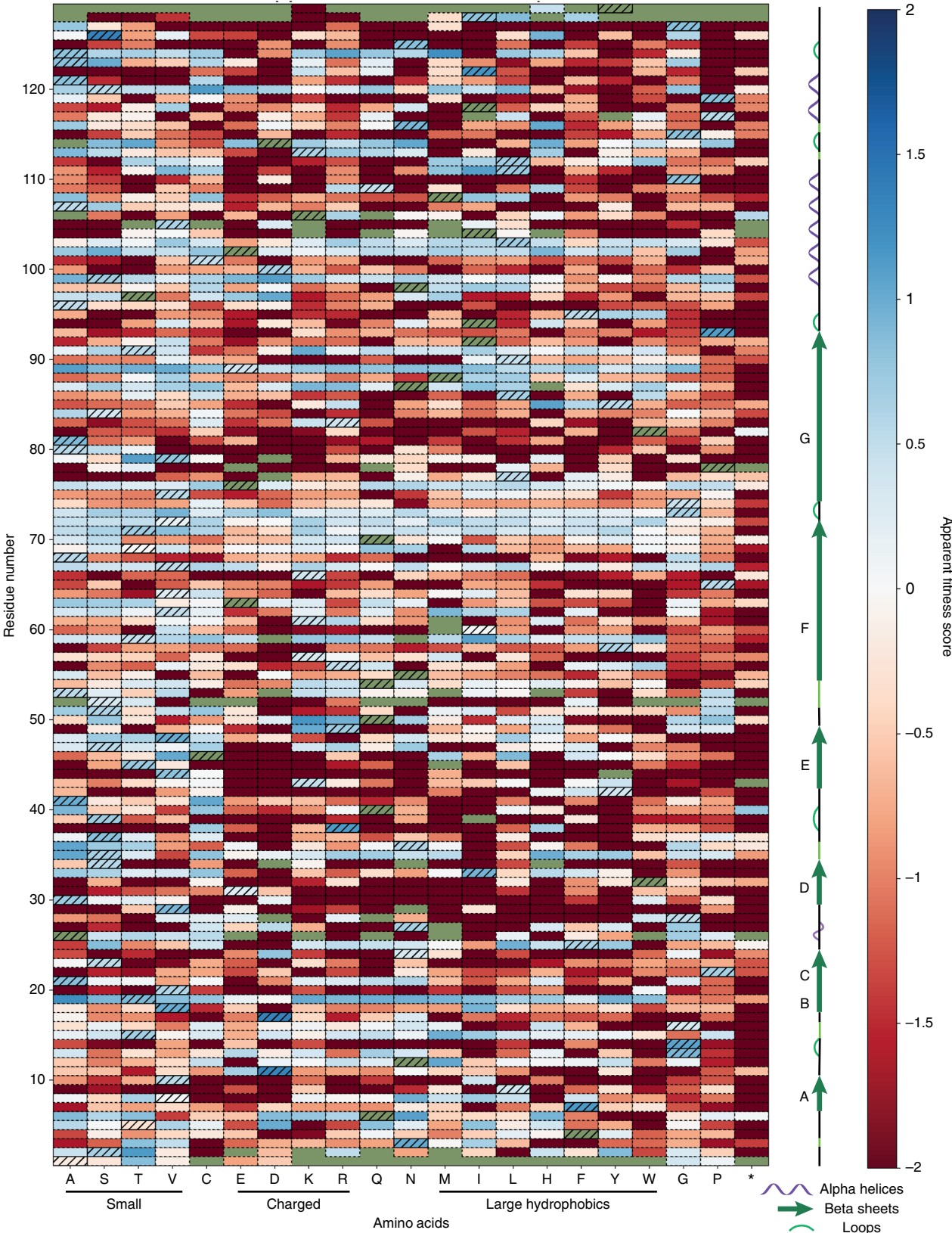

**Fig. 2** Apparent fitness scores (AFS, $n = 3$) for all single amino acid variants of the MS2 coat protein (MS2 CP). Wild-type residues are indicated with hatches, and missing values are green. Dark red variants were sequenced before selection but absent following selection

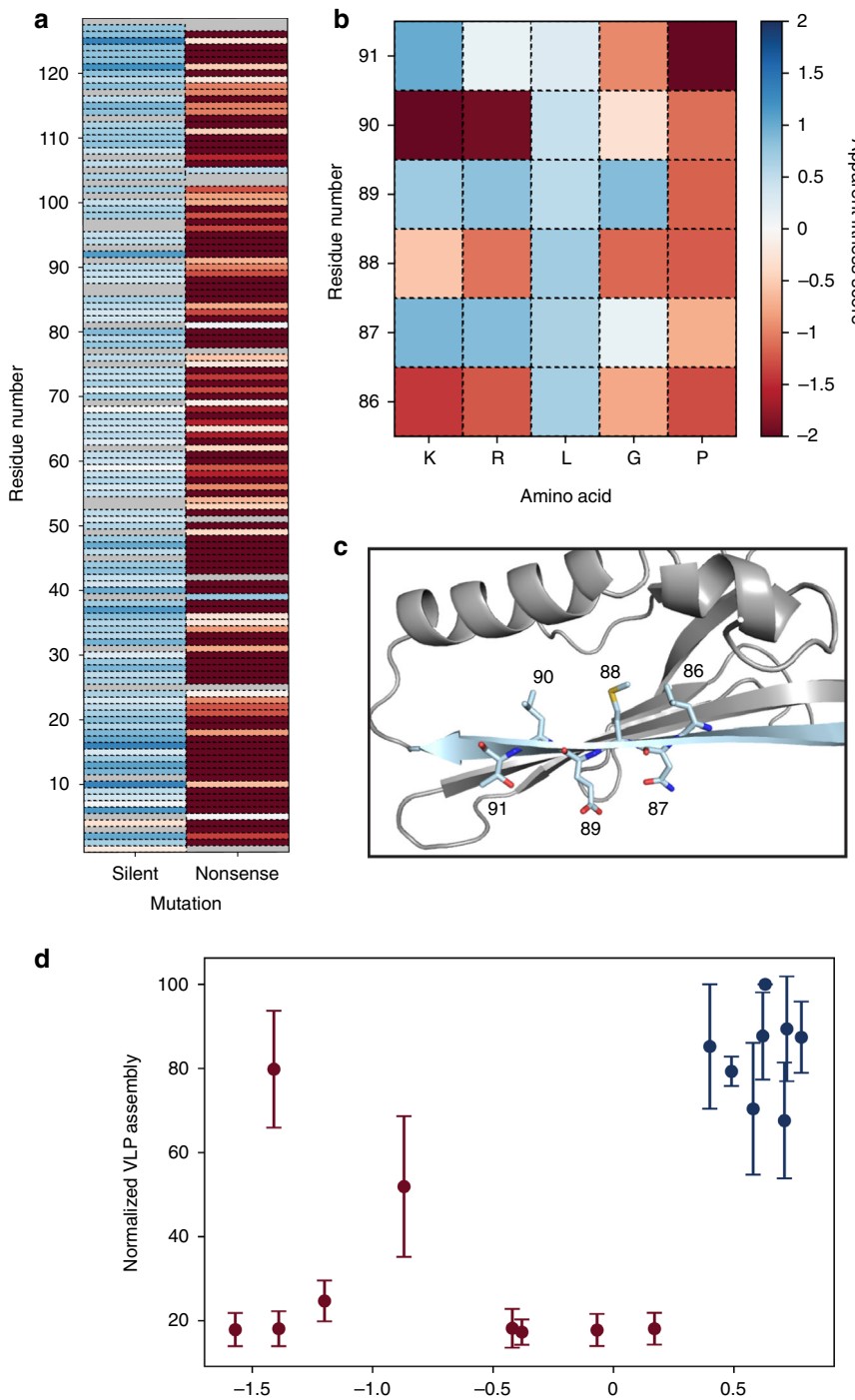

**Fig. 3** Validation of apparent fitness landscape (AFL). **a** Synonymous and nonsense mutation values are plotted against residue number. **b**, **c** Beta sheet G (light blue) shows predicted mutability patterns. **d** The assembly assay ($n = 3$) correlates well with AFS values. Red points were identified as non-assembling in the AFL, and blue points represent mutants with favorable assembly. Error bars indicate standard deviation

dimer structure. CP[S99P] inserts a proline at the base of an alpha helix. We also selected six cysteine variants patterned across the CP. In addition, several selected variants (CP[Q6V] and CP[Q50V]) have low AFS, though the residue is highly mutable, and one (CP [N24D]) has a high AFS at a poorly mutable residue.

All sixteen variants were characterized in an assembly assay (Fig. 3d). Fourteen of the sixteen variants showed similar trends when comparing apparent fitness to the assembly assay. False positives—that is, variants expected to assemble that do not—

would be problematic for applying fitness landscapes to engineering applications, but were not observed among the tested subset.

The assembly assay identified two variants that formed VLPs even though their AFS was low (CP[T91C] AFS = −1.4; CP [Q50C] AFS = −0.87). Upon further analysis by transmission electron microscopy (TEM), both false negatives (CP[T91C] and CP[Q50C]) were observed to form non-VLP, fibril-like aggregates in addition to wild-type-shaped VLPs (Supplementary Fig. 4). Several other variants (CP[P78L] and CP[N36C]) also formed

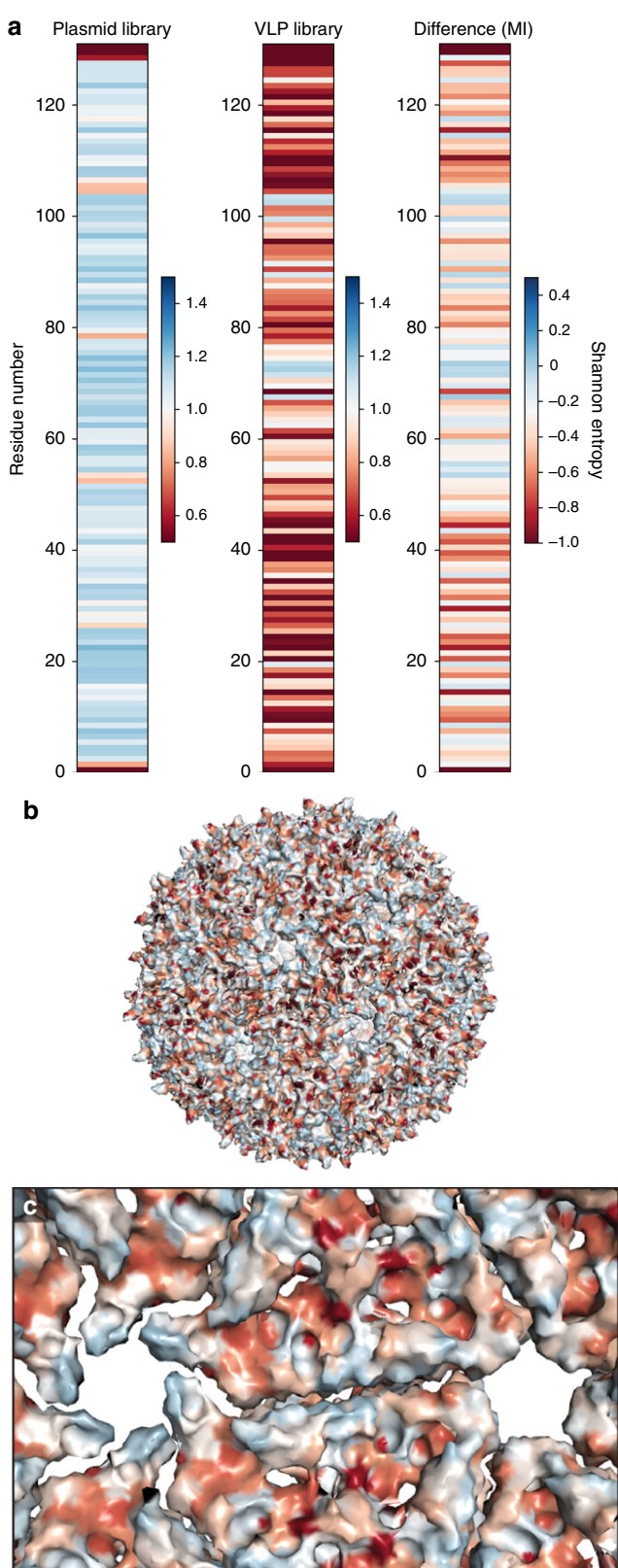

**Fig. 4** Mutability of MS2 CP. **a** The Shannon entropy (SE) of the plasmid library and VLP library are shown, along with the Mutability index (MI, VLP SE—plasmid SE). **b** The MI is shown mapped onto assembled MS2 CP (PDB ID = 2MS2) using the same coloring scheme as part A. **c** The MI is also shown from the interior perspective of assembled MS2 CP

non-VLP aggregates, though these did not form wild-type-shaped VLPs, and their AFS was low.

A similar fibril-like phenotype has previously been reported when high concentrations of the VLP assembly inducing oligonucleotide TR-DNA were applied to MS2 CP dimers during an in vitro reassembly assay[36]. TR-DNA induces a conformational switch from the C/C to A/B type dimers[57], and in this reported case, high concentrations of TR-DNA likely resulted in a dimer imbalance that produced non-VLP aggregates. Here, we hypothesize that these CP[T91C] and CP[Q50C] variations resulted in a similar imbalance in abundance of C/C or A/B type conformations, yielding the rod-like or fibril-like structures. These non-VLP aggregates likely were not enriched in our size-based selection, yielding a low AFS value. This result shows that variants forming non-VLP structures may be penalized in our selection.

**Interpretation of AFS**. In this work, we interpret variants with a negative AFS value as corresponding to capsids that exhibit low expression, assemble poorly, and/or are unstable toward protein purification conditions. All of these properties would likely limit their utility for delivery applications. As with any selection, the resulting quantitative fitness score could also be influenced by limited growth of the host cells, although this selection strategy eliminated attachment, whole-genome encapsulation, among other variables required for infectivity selections.

Of particular concern was whether mutations to the RNA binding pocket[58,59] in the VLP interior could yield a stable VLP that poorly binds RNA and thus is not detected by high-throughput sequencing. While we cannot rule out this possibility completely, we think it is unlikely for several reasons: (1) We are relying on passive nucleic acid encapsulation[38] rather than the specific and high-affinity interaction between the MS2 CP and the TR-RNA stem loop; (2) Simple negative charge is sufficient to stimulate VLP reassembly in vitro; (3) Recent structural data suggest many contacts between the MS2 CP and its genomic RNA[33], making individual point mutations less likely to influence the overall avidity of binding interactions; and (4) In the SyMAPS dataset, critical RNA-binding residues on average have high MI scores. Indeed, of the eleven RNA binding residues, only two fell lower than the 30th percentile in mutability (Supplementary Fig. 5). An additional three residues had mutability scores between the 30th and 70th percentile, and the remaining six had a MI higher than 70% of other residues.

Sequencing read errors are the most likely reason to generate false positives; indeed, two of the 129 nonsense mutations have AFS values higher than zero, even though nonsense mutations cannot form well-formed VLPs. Upon closer inspection, both of the wild-type codons at these positions are one base pair away from TAG, the stop codon present in an NNK codon (residue 40, CAG; residue 106, AAG), making these positions more susceptible to sequencing read errors. We generated a higher stringency AFL that only analyzed codons that are two or more base pairs away from wild type, essentially eliminating the effect of sequencing read errors (Supplementary Fig. 6). We find that mutability trends hold in the stringent AFL, leading us to conclude that sequencing read errors minimally affect the AFL.

**AFL shows complexity of self-assembly**. Ten physical properties of amino acids were evaluated to determine how each property contributes to mutability across the VLP (Fig. 5). Size-based parameters (volume, molecular weight, length, and steric bulk) seem to group by region, where a set of several proximally positioned residues show similar preferences. In contrast, polarity, polar area, and flexibility show a distinct banding patterns

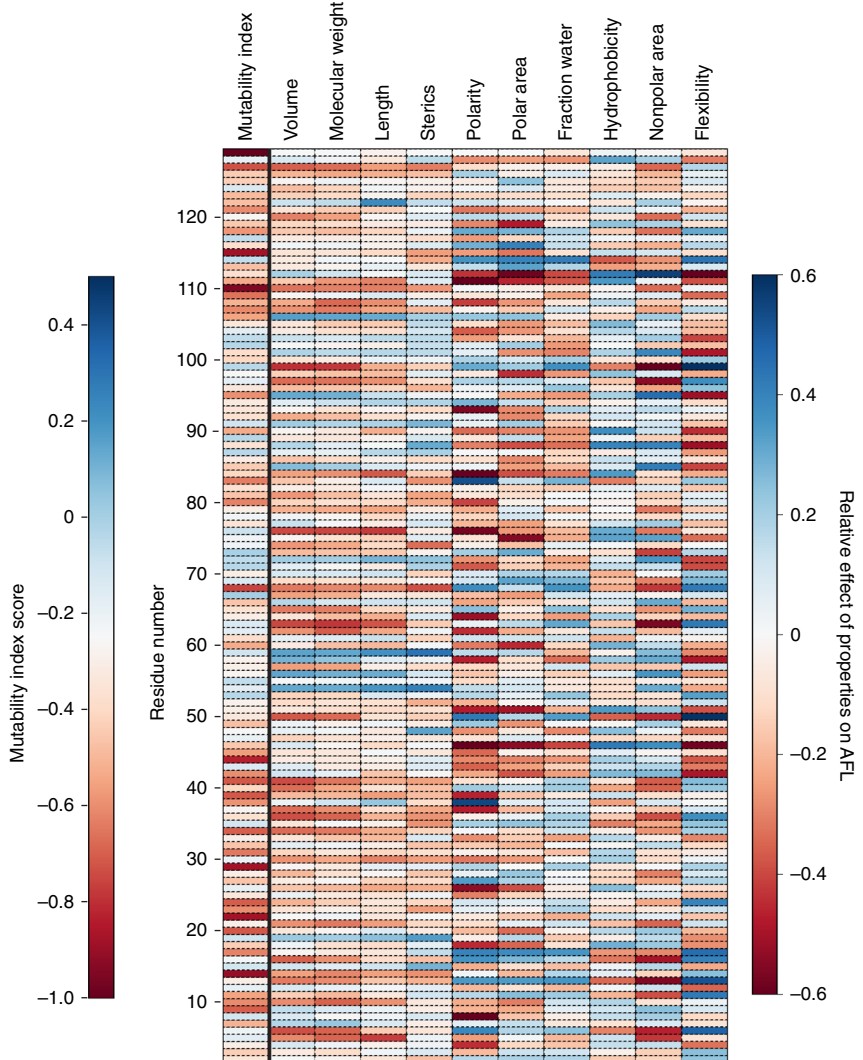

**Fig. 5** The effect of physical properties on the apparent fitness landscape (AFL). Each value represents the sum of the apparent fitness score (AFS) multiplied by a scalar corresponding to the indicated property. Blue scores mean that the position showed a preference for high property values, while red scores show a preference against high property values. Mutability index is shown as a reference

across the protein, indicating a strong preference based on the local environment of a single residue.

This analysis can also be used to understand the physical preferences of an individual residue. For example, residue S99 is a relatively mutable residue at the base of an alpha helix. We can see that position 99 prefers smaller, flexible, or polar residues, but is biased against hydrophobic residues. In contrast, position E89—one of the most mutable residues in the VLP—has minimal preferences for any particular property except flexibility, which is disfavored.

We can further use these analyses to identify locations to insert non-native amino acids, which tend to be larger than native amino acids. Several residues seem to prefer larger amino acids, including residue T19, a position where non-native residues have been installed. Combining these analyses with the MI and AFL, we anticipate that residues Q54 or T59 may be additional locations where we can install non-native amino acids. Taken together, the complexity of these results highlights the importance of measuring a fitness landscape directly.

**Engineering an acid-sensitive MS2 CP variant**. We hypothesized that variants in this library may have useful differences in physical properties compared to the wild-type MS2 CP. To uncover specific

variants that exhibit desired traits, we applied additional selective pressures to the library. MS2 VLPs have been used to deliver a variety of cargo to cells other than the canonical *E. coli* hosts, including sensitive biomolecules, such as proteins and RNA[35,36]. In these cells, release from the VLP is presumed to occur via an acid-triggered cargo release mechanism in the late endosome or lysosome[35]. However, an in vitro acid screen revealed that the CP [WT] maintains ~90% soluble VLP at the acidity of lysosomes (pH 4.5). We reasoned that delivery applications could benefit from an MS2 CP variant that is less tolerant to these acidic conditions.

We exposed the library of MS2 CP variants to pH 5 at physiological temperature (37 °C) for four hours, mimicking the conditions of a human endosome or lysosome. Under these conditions, two variants (CP[T71H] and CP[E76C]) were predicted to form particles but were found to have a lower percent abundance following the selective pressure (Supplementary Fig. 7a). Other possible hits were eliminated either because the variant was not predicted to form assembled VLPs (CP [A41G]) or was predicted to be temperature sensitive rather than acid-sensitive (CP[T71R]).

We constructed both variants and conducted an in vitro acid screen (Fig. 6). Indeed, CP[T71H] exhibited a reduction in well-

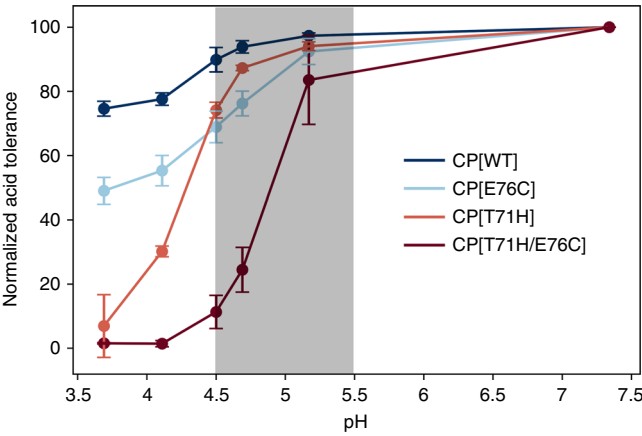

**Fig. 6** Reduced acid tolerance of MS2 CP[T71H] and CP[T71H/E76C]. Both CP[T71H] and CP[T71/E76C] are less acid tolerant than CP[WT] and CP [E76C]. Normalized acid tolerance ($n = 3$) indicates HPLC SEC peak height relative to pH 7.3. The highlighted region is ideal for endosomal disassembly

formed particles between pH 4.5 and 3.5 compared to CP[WT], confirming that we successfully identified a variant that is less stable to mildly acidic conditions than the CP[WT] using SyMAPS. At this pH range—which is near the acidity of late lysosomes—we observed aggregation and precipitation of CP [T71H] (Supplementary Fig. 7b-c), and these precipitated VLPs were morphologically irregular by TEM. We then combined CP [T71H] with CP[E76C] to form a double mutant that contained both predicted acid-sensitive variants. The acid sensitivity effect was additive, yielding even more acid sensitive variants that aggregated near pH 5. However, expression yields and storage stability were compromised, suggesting that additional rounds of optimization are still needed.

We hypothesize that the decreased stability of the protein cage in mildly acidic conditions could enhance the rate of cargo release in endosomes and lysosomes. Thus, we predict that CP[T71H] or an optimized double mutant may improve endosomal release without compromising the circulation stability and cargo protection properties that make the MS2 CP an ideal drug delivery vehicle.

## Discussion

Many patterns observed in our data challenge conventional protein engineering assumptions. Smaller or hydrophobic amino acids, including valine and, surprisingly, glycine, were among the best-tolerated amino acids across the CP and may be useful to remove undesirable chemical functionalities (Fig. 7a). Negative charges, bulky residues, and proline were poorly tolerated across the entire gene, and nonsense mutations were still much more detrimental, as expected. Overall, we see that the structure is more immutable than mutable, though a surprising number of locations tolerated modifiable amino acids like cysteine and lysine. This may be useful for increasing the capacity of the MS2 CP to carry small molecule cargo.

Several conventional protein engineering techniques are directly evaluated in this system. For example, alanine scanning is often used to identify amino acids that contribute to stability or function in a protein, as alanine is assumed to be a neutral mutation for most native residues. To evaluate this assumption, we plotted the AFS value for alanine compared to every other amino acid, then calculated the Pearson correlation for each scenario. Our results indicate that, surprisingly, alanine shows minimal correlation in mutability to most other amino acids and shows particularly poor or negative correlations to bulky amino

acids, such as phenylalanine and tryptophan (Supplementary Fig. 8a). Indeed, our results show that alanine scanning may artificially inflate the chances of identifying a bulky amino acid as critical to protein stability, as the mutation likely is not neutral.

We used the same correlations to show that while lysine and arginine AFS values correlate well, and glutamate and aspartate AFS values correlate well, the two types of charges correlate poorly to one another (Supplementary Fig. 8b). This result indicates that positively and negatively charged residues, which are often evaluated together, actually behave quite differently in this system. In addition, we see more nuanced variation in substitutability between valine, isoleucine, and leucine, three hydrophobic residues that typically are considered quite similar. While isoleucine AFS values correlate well with valine and leucine AFS values, valine and leucine AFS values correlate less well with one another (Supplementary Fig. 8c), indicating that the position of the methyl branch is likely important in determining amino acid substitution patterns.

The influence of the native amino acid on substitution at a particular residue in the MS2 CP was evaluated (Fig. 7b). Substituted residues of similar chemistries cluster together, as would be expected. The MS2 CP has ten native positively charged residues and nine native negatively charged residues. On average, negatively charged residues permitted more non-native substitutions than positively charged residues. Several residues did not tolerate any single amino acid well, including glycine (nine native residues), proline (six native residues), and leucine (seven native residues). Surprisingly, threonine (nine native residues) tolerated many substitutions, perhaps because it is both bulky and hydrophilic in nature. While only two cysteines are present in the native sequence, it is interesting to note that serine was not tolerated as a mutation for either cysteine. One cysteine did not tolerate any mutations, while the other tolerated residues with similar hydrophobicity index to cysteine, such as valine and leucine. No one substituted amino acid was well tolerated at every native amino acid.

In one step, we recapitulated and expanded on many studies that characterized where mutations can be installed in the MS2 CP, both on the interior and exterior of the delivery vehicle. In this study, 530 MS2 variants were characterize that permit MS2 VLP formation. Included in this number are 35 cysteine mutants and 28 lysine mutants, which indicates that additional reactive groups may be able to be installed to increase the modification rates and carrying capacity of the MS2 CP as a delivery vehicle. In addition to selecting for assembly alone, we selected for a variant that is less tolerant to acidic conditions than wild type. This previously unknown variant, CP[T71H], is less tolerant to pH 4, which is near the pH of late endosomes or early lysosomes. Moving forward, we and others can use AFL and MI values generated in this study to guide where mutations of interest can be installed in the MS2 CP. These results may also be directly applicable to identify mutable residues in structurally related VLPs, such as Qβ. This map saves significant time and effort and allows researchers to produce MS2 CP variants with rationally engineered, highly tunable chemical or physical properties.

## Methods

**Entry vector generation (EMPIRIC cloning).** To generate libraries with single amino acid mutations, we modified a cassette ligation strategy developed by the Bolon lab in 2011[48]. This strategy uses a plasmid with self-encoded removable fragments (SERF) that are surrounded by inverted BsaI restriction sites, so BsaI digestion removes both the SERF and BsaI sites. These plasmids will be referred to as entry vectors, and the SERF contains constitutively expressed GFP to enable green/white screening. We divided the MS2 CP into five segments of 26 codons in length—a decision that allowed us to mutagenize the MS2 CP by purchasing 100 basepair single-stranded DNA primers rather than full-length double-stranded DNA, which can be more expensive. Five golden gate compatible entry vectors were synthesized with constitutively-active GFP swapped for a 26-codon segment of the MS2 gene[48,60] (Supplementary Data 4). Inverted BsaI cut sites were then introduced into the vectors using QuikChange mutagenesis[61].

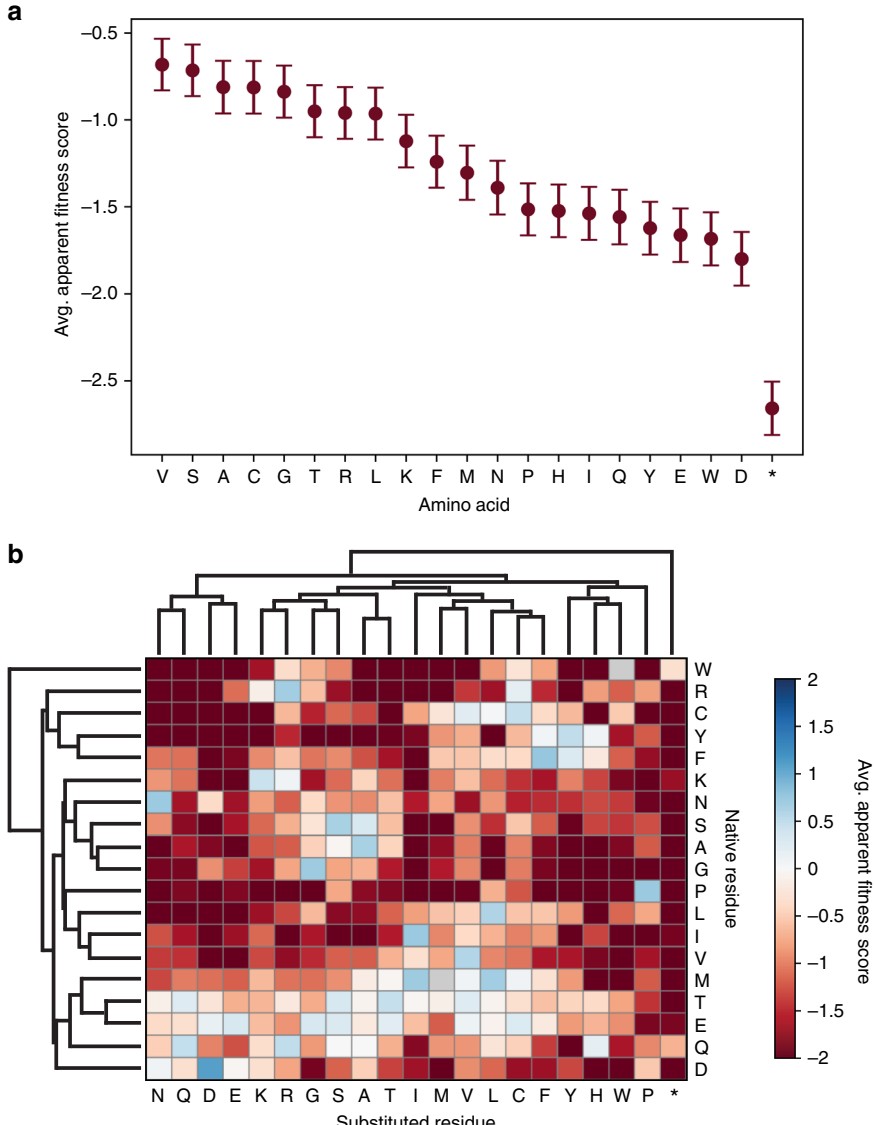

**Fig. 7** Average AFS values ($n = 3$) of amino acids across MS2 CP. **a** The average AFS and standard error values are indicated for every native residue mutated to the indicated amino acid. **b** Averages from part A are separated by native residue and clustered by similarity

**CP variant library generation (EMPIRIC cloning)**. Single-stranded DNA primers were purchased that spanned the length of each 26-codon region of MS2 with appropriate cut sites for each corresponding entry vector (Supplementary Data 4). Primers for each entry vector were resuspended, pooled, and diluted to a final concentration of 50 ng/μL. The reverse strand was filled in using a touchdown PCR[62] with 10-mers directed to the golden gate cut sites. The amplified, double-stranded DNA was purified using a PCR Clean-up Kit (Promega, Cat# A9282), then diluted to 1–5 ng/μL. These mixtures were cloned into their respective entry vectors using EMPIRIC cloning[48], which relies on established golden gate cloning techniques[60]. The ligated plasmids were transformed into chemically competent DH10B *E. coli* and plated on large ($245 \times 245 \times 20$ mm, #7200134, Fisher) LB-A plates with 32 μg/mL chloramphenicol. Colony number varied, but every transformation yielded a number of colonies that was at least three times the theoretical library size. This protocol was repeated in full for three total biological replicates that are fully independent from library generation through selection.

**CP variant library expression and purification**. Colonies were scraped from the plates, combined into LB, and allowed to grow for 2 h. The pools of variants were combined by OD600 into 1 L of 2xYT (Teknova, Cat: Y0210) to generate a library of CP variants. These variants were allowed to grow to an OD of 0.6, when they were induced with 0.1% arabinose. Expression proceed overnight at 37 °C, and cultures were harvested and sonicated. Two rounds of ammonium sulfate precipitation at 50% saturation were followed by fast protein liquid chromatography (FPLC) size exclusion chromatography (SEC) purification (Supplementary Fig. 9a). Fractions 12–22 were harvested (Supplementary Fig. 9b).

**Sample preparation for high-throughput sequencing**. Plasmid DNA was extracted prior to the assembly selection using a Zyppy Plasmid Miniprep Kit (Zymo, Cat# D4036). RNA was extracted from the CP library after the assembly selection following previously published protocols[63]. TRIzol (Thermo Fisher Cat# 15596026) was used to homogenize samples, followed by addition of chloroform. The sample was separated into aqueous, interphase, and organic layers. The aqueous layer (containing RNA) was isolated, and the RNA was precipitated with isopropanol and washed with 70% ethanol. RNA was then briefly dried and resuspended in RNAse free water. cDNA was synthesized using the Superscript III first strand cDNA synthesis kit from Life (cat: 18080051, polyT primer). cDNA and plasmids were then amplified with two rounds of PCR to add barcodes (10 cycles) and the Illumina sequencing handles (8 cycles), following Illumina 16S metagenomic sequencing library preparation recommendations (Supplementary Data 4). Libraries were combined and analyzed by 300 PE MiSeq in collaboration with the UC Davis Sequencing Facilities. 20.6 million reads passed filter, and an overall Q30 > 79%.

**FPLC SEC (SyMAPS size selection)**. MS2 libraries and mutants were purified on an Akta Pure 25 L FPLC system with a HiPrep Sephacryl S-500 HR column (GE Healthcare Life Sciences, Cat# 28935607), SEC column via isocratic flow with 10 mM phosphate pH 7.2, 200 mM sodium chloride, and 2 mM sodium azide. Fractions containing MS2 coat protein were harvested. FPLC SEC traces of SyMAPS replicate 1, 2, and 3, and their comparison to the wild-type CP protein, can be found in Supplementary Fig. 9.

**FPLC anion exchange**. Individual CP variants were purified on an Akta Pure 25 L with a hand-packed DEAE Sepharose Fast Flow column (GE Healthcare Life Sciences, Cat# 17070901). Variants were eluted with 20 mM taurine pH 9.

**HPLC SEC**. MS2 CP variants were analyzed via isocratic flow on an Agilent 1290 Infinity HPLC with an Agilent Bio SEC-5 column (5 um, 2000 A, 7.8 × 300 nm) with isocratic flow with 10 mM phosphate pH 7.2, 200 mM sodium chloride, and 2 mM sodium azide. Wild-type MS2 has a characteristic elution peak at 11.2 min, and peak height was used as a proxy for VLP formation.

**Transmission electron microscopy**. Samples were prepared for TEM analysis by applying an analyte solution (A280 of approximately one) to carbon-coated copper grids for 2 min, followed by triple rinsing with dd-$H_2O$. The grids were then exposed to a 1.6% aqueous solution of uranyl acetate for 1 min as a negative stain. Images were obtained at the Berkeley electron microscope lab using a FEI Tecnai 12 transmission electron microscope with 120 kV accelerating voltage.

**Agarose gel electrophoresis**. PCR products were analyzed in a 1% agarose gel in TAE buffer (40 mM Tris, 20 mM acetic acid, and 1 mM EDTA) with 2X SYBR Safe DNA Gel Stain (ThermoFisher Scientific, Cat# S33102) for 30 min at 120 volts. Agarose gels were imaged on a BioRad GelDoc EZ Imager.

**SDS-PAGE analysis**. NuPAGE 4–12% Bis-Tris Protein Gels (Invitrogen, Cat# NP0323BOX) were used. Gels were run with 1X MES buffer for 45 min at 160 volts. Samples were loaded with Laemmli sample buffer and imaged with a Coomassie stain on a BioRad GelDoc EZ Imager.

**Strains**. *Escherichia coli* DH10B competent Cells from ThermoFisher Scientific were used for all experiments. Overnight growth from a single colony was grown for 16–20 h at 37 °C shaking at 200 r.p.m. in LB-Lennox media (VWR, Cat# AAH26760) with chloramphenicol at 34 mg/L. Expressions were sub-cultured 1:100 into 2xYT media (Teknova, Cat# Y0210) with 34 mg/L chloramphenicol and allowed to express overnight at 37 °C shaking at 200 r.p.m.

**Individual variant cloning**. Individual variants were cloned using a method adapted from above. Briefly, overlap extension PCR (Supplementary Data 4) yielded a double-stranded fragment that spanned the length of one entry vector chunk. Each fragment was cloned into its respective entry vector using standard golden gate cloning techniques. Cloned plasmids were transformed into DH10B cells. Individual colonies were sequenced prior to expression.

**Individual variant assembly-competency screen**. Sixteen selected mutants were individually then expressed in 50 mL cultures of 2xYT as described. These expressions were lysed by sonication, precipitated twice with 50% ammonium sulfate, and evaluated by HPLC, SEC, and TEM. Peak height percent compared to wild type at 11.2 min was used as a proxy for VLP formation.

**Individual variant expression and purification**. Individual variants identified as potentially acid sensitive were expressed as described previously, precipitated with 50% ammonium sulfate, and purified with FPLC anion exchange as described above. Fractions containing MS2 CP variants were buffer exchanged into 10 mM phosphate pH 7.2 with an Amicon Ultra-15 Centrifugal Filter with a 100 kDa membrane filter. Concentrated variants were frozen for further analysis.

**High-throughput sequencing data processing**. Data were trimmed using Trimmomatic[64] with a 2-unit sliding quality window of 30 and a minimum length of 30.

*java -jar trimmomatic-0.36.jar PE input_forward_HTS001.fastq.gz input_reverse_HTS001.fastq.gz s1_pe s1_se s2_pe s2_se SLIDINGWINDOW:2:30 MINLEN:30*

Reads were merged with FLASH (fast length adjustment of short reads)[65] with a maximum over-lap of 167 basepairs.

*flash -M 167 s1_pe s2_pe -o HTS001*

Reads were then aligned to the wild-type MS2 CP reference gene with Burrows–Wheeler Aligner (BWA-MEM)[66].

*bwa mem -p Reference/ref.fasta HTS001.extendedFrags.fastq>HTS001.sam*

Reads were sorted and indexed with Samtools[67].

*samtools view -bT Reference/ref.fasta HTS001.sam -o HTS001.bam samtools sort -o HTS001_sort.bam HTS001.bam*

*samtools index HTS001_sort.bam*

The Picard function CleanSam was used to filter unmapped reads.

*java -Xmx4g -jar picard.jar CleanSam I=HTS001_sort.bam O=HTS001_filt.bam*

Reads longer or shorter than the expected length of the MS2 CP were removed.

*samtools view -b -F 4 HTS001_filt.bam>HTS001_map.bam*

*samtools view HTS001_map.bam | grep "393M" | sort | less -S>HTS001.txt*

Reads were proceed to generate an AFL using code written in-house (Supplementary Methods).

**Code availability**. All in-house code is available from the authors upon request. Additional details on computational methodology can be found in the Supplementary Information.

**Data availability**. AFL and MI data for the MS2 CP are available in the Supplementary materials, and additional information is available from the authors by request.

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

## Acknowledgements

This work was funded by the Army Research Office (W911NF-15-1-0144 and W911NF-16-1-0169) and the BASF CARA program at UC Berkeley. E.C.H. was supported under by the DoD, Air Force Office of Scientific Research, National Defense Science and Engineering Graduate (NDSEG) Fellowship, 32 CFR 168a. E.A.B. was supported under the NIH-MARC Program undergraduate fellowship (5T34GM007821-38) and the Amgen Scholars Foundation. We would like to thank Dr. Ke Bi in the Computational Genomics Research Facility at UC Berkeley and Han Teng Wong for helpful discussions. The sequencing was carried by the DNA Technologies and Expression Analysis Cores at the UC Davis Genome Center, supported by NIH Shared Instrumentation Grant S10 OD010786.

## Author contributions

E.C.H., C.M.J., M.B.F., and D.T.E. conceived this project. E.C.H. and E.A.B. performed experiments for this project, and E.C.H., C.M.J., A.H.F., and M.J.L. analyzed the AFL. E.C.H., M.B.F., and D.T.E. wrote the manuscript. All of the authors reviewed and contributed to the manuscript.

## Additional information

**Competing interests:** The authors declare no competing interests.

