## [Peer Review File(PDF 243 kb) · Nature Communications]

Reviewers' comments:

Reviewer #1 (Remarks to the Author):

In this study, authors developed a selection and calculation strategy for quantitative description about the MS2 viral capsid assembly. The study elegantly combined the experimental and computational method to produce useful and predictable information about the viral assembly, with potential application to other viral and protein/peptide assemblies. This reviewer is satisfied with the overall significance and results of the research; however, there are the main drawback of this manuscript in that the description about the system and method is rather cryptic. This reviewer strongly urges that the manuscript needs substantial revision to provide more detailed description about the system.

- Many information is missing. Detailed experimental procedures and especially calculation methods are not well described. Without that, it would be difficult for others to replicate or apply similar methods to other system.
- What is Shannon entropy? Is there any reference? How it is calculated?
- Figure 2: Why S and T are classified as hydrophobic residues? They are usually classified as polar and hydrophilic residues.
- Figure 4: It would be informative to other researchers to provide specific MI scores for each residue, probably in the SI.

Reviewer #2 (Remarks to the Author):

The overall idea and outcome of the approach presented in this paper is novel, very interesting and is potentially a powerful approach to provide useful information for engineering VLPs and proteins in general. However, it seems to me that there are a few critical points, which were not discussed in the manuscript, or presented in a way that could be misleading.

L86: It is unclear whether the genotype-to-phenotype link referred to is MS2's ability to encapsulate RNA, to self-assemble, or both.

A concern I have is effect of point mutation on binding affinity to genomic RNA, which potentially biases the interpretation of the outcome data. They sequenced library of MS2 variants at two different stages. 1) sequenced an established plasmid library before transforming to E. coli. 2) extracted genomic RNAs from expressed VLP library, reverse transcribed them to cDNAs and sequenced. Identification (and estimation of relative abundance) of assembled VLPs is totally dependent on ssRNA encapsulated in each VLP variant. If a point mutation diminishes or destroys RNA binding capability, it seems that the apparent fitness score (AFS) could be underestimated. In the worst case scenario, a VLP variant could be undetected although it assembles as well as wtVLP, if the variant lost its RNA binding capability. If so, this should be mentioned/discussed in the manuscript as a limitation of the presented approach.

Needless to say, CP - ssRNA interaction of MS2 has been well studied and several amino acid residues of MS2 CP, which are involved in RNA binding, have been identified (eg by Peter Stockley's group and David Peaboy's group). I think that the authors should discuss their results in this context. For example some of the amino acids involved in RNA binding are 45Thr, and 59Thr (according to Stockley's paper). According to Fig 2 and Fig 5, 45Thr shows quite low (negative) "mutability index score", whereas 59Thr shows relatively high (positive) "mutability index score". Does this mean 45Thr could be more critical for RNA binding than 59Thr?

In this regard, Figure 4C ("mutability index map of interior surface") might be misleading, because

deep red on this map does not necessarily suggest "no assembly to a proper VLP" but could be "assembled into VLP but lack of RNA", (so these sites are mutable). Where in this map is RNA binding site?

I was surprised that the effects of point mutations on RNA binding was not discussed in the manuscript because it seems such an important point to me – perhaps I have missed something...?

L137-138: It is not clear where the diamond-shaped pattern appears in Figure 4c or why it is significant.

L165-166: Describe how residues were selected randomly. 4 proline and 6 cysteine residues out of 16 total residues does not seem random.

L246-L251: The authors claimed that "serine was not often tolerated as a mutation for cysteine" and "instead, valine and leucine were better substituted for native cysteine residues". However, there are only two native cysteine in MS2 CP. Furthermore, according to Fig 2, one of them (46Cys) can be substituted to valine or leucine, but the other (101Cys) can not. Therefore, it seems to me that it is difficult to generalize solely based on the data presented in this paper. Is this discussion based on not only the data presented in this manuscript but also previous protein fitness landscape studies done by others? Is this claim consistent with previous studies? This is not clear from the manuscript.

Method L 284: "This protocol was repeated in full for three total biological replicates". It is not clear to me if it is 1) three totally independent replication, or 2) three consecutive cycle (like phage display library) i.e. use cDNA library of the 1st cycle to generate plasmid library of the 2nd cycle to narrow down toward more mutable variants.

(Judging from other part of the manuscript, the first one is probably the case but I am not 100% sure and it would be a good idea to be explicit on this point)

Experimental procedure of a size based selection process is not well described in either main text or supplement. It would be helpful to understand a procedure if a SEC profile of VLP mixture is presented and explained what fractions are considered as "assembled" VLP, in comparison with SEC profile of wtVLP.

Fig 2 and 3: There are two nonsense mutations which have positive fitness score. How is this possible?

Other Minor Points

A more in depth discussion of protein fitness landscapes would be useful and a brief explanation of the modified EMPIRIC cloning method used.

Most of the first paragraph of the results section is an extension of the introduction. I would suggest removing it from the results section.

L80: A statement of ".... stable towards harsh condition." is a little vague. It would be better to describe more e.g. temperature and pH range etc.

Figure 1: would benefit from a more in depth caption.

Figure 2: It is very difficult to distinguish gray color (indicating missing mutations) from blue, particularly pale water color blue. Would be a good idea to use a different color or pattern for missing mutations. e.g. black, or mesh.

L243-245: It is mentioned that "the two branched residues (I/V) behave more similarly than the two isomers (I/L)". However, Leu also has "branched" side chain like Val and Ile.

"AFL" in L214, L261, and Figure 5 right side legend: Are these a typo of AFS, or does this mean something else?

Figure 5: There is a typo in the word flexibility (top right).

Reviewer #3 (Remarks to the Author):

The manuscript provides a comprehensive description of mutations on coat protein that impact the assembly of the MS2 capsid. The study design is straightforward and the experiment was executed properly. However the novelty is limited in the current version.

Many studies have presented comprehensive fitness mapping of proteins or viral genomes in the past several years with the advancement of sequencing capacity. In this study, the selection is size exclusion chromatography. However the method cannot separate viral capsids from aggregates of similar sizes, which creates background noise. About 75% of targeted variants were observed in the study, an acceptable coverage, but not ideal. Potential explanations for the phenotypes were provided on the basis of biophysical properties of amino acids, consistent with expectations. But the "Apparent fitness score predicts VLP assembly" is an over-statement. It is verification not prediction. The potential impact on drug delivery is quite distant. If the authors can engineer new capsids with desired properties using the comprehensive fitness map, the novelty and impact of the study will be significantly elevated.

Reviewer: 1

1. *Many information is missing. Detailed experimental procedures and especially calculation methods are not well described. Without that, it would be difficult for others to replicate or apply similar methods to other system.*

We apologize for these omissions. We added extensive experimental procedures and computational methods to the supplemental information to ensure that others can replicate these results. Included in these additions is line-by-line explanation of how data were processed at every step along the analysis pipeline.

2. *What is Shannon entropy? Is there any reference? How it is calculated?*

We have added a line to the supplemental information to describe how Shannon Entropy is calculated. We additionally added the following reference to line 150 to support how Shannon Entropy can be used as a proxy for mutability:

Stewart, J. J., Lee, C. Y., Ibrahim, S., Watts, P., Shlomchik, M., Weigert, M. & Litwin, S. A Shannon entropy analysis of immunoglobulin and T cell receptor. *Mol. Immunol.* **34**, 1067–1082 (1997).

3. *Figure 2: Why S and T are classified as hydrophobic residues? They are usually classified as polar and hydrophilic residues.*

We apologize for this mistake. We changed the label from “small hydrophobic” to “small” in all cases, and cysteine was removed from the group.

4. *Figure 4: It would be informative to other researchers to provide specific MI scores for each residue, probably in the SI.*

We agree with the reviewer that this will be helpful to other researchers. We added a CSV file containing the Mutability Index scores to the supplemental files. In addition, we also added a second CSV file containing the Apparent Fitness Landscape to the supplemental files.

Reviewer: 2

The overall idea and outcome of the approach presented in this paper is novel, very interesting and is potentially a powerful approach to provide useful information for engineering VLPs and proteins in general. However, it seems to me that there are a few critical points, which were not discussed in the manuscript, or presented in a way that could be misleading.

1. *R2: L86: It is unclear whether the genotype-to-phenotype link referred to is MS2's ability to encapsulate RNA, to self-assemble, or both.*

We apologize for the confusion. We add several sentences to clarify the genotype-to-phenotype link used in this selection [lines 98–105], which is also detailed in response to comment 2, below.

- 2. A concern I have is effect of point mutation on binding affinity to genomic RNA, which potentially biases the interpretation of the outcome data. They sequenced library of MS2 variants at two different stages. 1) sequenced an established plasmid library before transforming to E. coli. 2) extracted genomic RNAs from expressed VLP library, reverse transcribed them to cDNAs and sequenced. Identification (and estimation of relative abundance) of assembled VLPs is totally dependent on ssRNA encapsulated in each VLP variant. If a point mutation diminishes or destroys RNA binding capability, it seems that the apparent fitness score (AFS) could be underestimated. In the worst case scenario, a VLP variant could be undetected although it assembles as well as wtVLP, if the variant lost its RNA binding capability. If so, this should be mentioned/discussed in the manuscript as a limitation of the presented approach.*

We apologize for the confusion and thank the reviewer for these comments. We addressed these concerns in several ways. First, we clarified that we are not relying on the well-studied, high-affinity stem loop TR-RNA–protein interaction that is commonly used with the MS2 CP [lines 98–101]. As the reviewer mentioned, the affinity for this stem loop to bind the MS CP can be altered by amino acid mutations. Instead, we are using passive encapsulation, where the MS2 CP nucleates on available negative charge, including the mRNA encoding the MS2 CP [lines 101–105]. This passive encapsulation gives us a snapshot into the mRNA available in a cell, but only if assembly occurs. Because this interaction relies on many lower affinity contacts, we anticipate that single amino acid substitutions will change the interaction less dramatically. We additionally added a paragraph discussing whether RNA binding could alter the Apparent Fitness Landscape [lines 224–235].

- 3. Needless to say, CP – ssRNA interaction of MS2 has been well studied and several amino acid residues of MS2 CP, which are involved in RNA binding, have been identified (eg by Peter Stockley's group and David Peaboy's group). I think that the authors should discuss their results in this context. For example some of the amino acids involved in RNA binding are 45Thr, and 59Thr (according to Stockley's paper). According to Fig 2 and Fig 5, 45Thr shows quite low (negative) "mutability index score", whereas 59Thr shows relatively high (positive) "mutability index score". Does this mean 45Thr could be more critical for RNA binding than 59Thr? In this regard, Figure 4C ("mutability index map of interior surface") might be misleading, because deep red on this map does not necessarily suggest "no assembly to a proper VLP" but could be "assembled into VLP but lack of RNA", (so these sites are mutable). Where in this map is RNA binding site? I was surprised that the effects of point mutations on RNA binding was not discussed in the manuscript because it seems such an important point to me – perhaps I have missed something...?*

We added a supplemental figure showing the mutability of the canonical RNA binding regions (Figure S5). In lines 231–235, we additionally discussed that many of these residues are highly mutable, leading us to believe that RNA binding is not dramatically altering the Apparent Fitness

Landscape. Because we are not using the well-studied ssRNA stem loop, we do not feel that we can draw conclusions about the relative importance of RNA binding residues from these studies.

4. *L137-138: It is not clear where the diamond-shaped pattern appears in Figure 4c or why it is significant.*

We removed this phrase from the paper. Additionally, we included the PyMol file with Mutability Indices mapped onto the capsid as a supplemental file so other researchers can more easily examine the mutability of the MS2 CP.

5. *L165-166: Describe how residues were selected randomly. 4 proline and 6 cysteine residues out of 16 total residues does not seem random.*

We apologize for this error. We changed this sentence to more accurately reflect the selection criteria [lines 186–188]. The residues were selected as examples and were not random.

6. *L246-L251: The authors claimed that "serine was not often tolerated as a mutation for cysteine" and "instead, valine and leucine were better substituted for native cysteine residues". However, there are only two native cysteine in MS2 CP. Furthermore, according to Fig 2, one of them (46Cys) can be substituted to valine or leucine, but the other (101Cys) can not. Therefore, it seems to me that it is difficult to generalize solely based on the data presented in this paper. Is this discussion based on not only the data presented in this manuscript but also previous protein fitness landscape studies done by others? Is this claim consistent with previous studies? This is not clear from the manuscript.*

We apologize for the confusion and thank the reviewer for this comment. We added detail to clarify that these discussions are solely within the context of the MS2 CP and not other proteins [lines 324–325]. In addition, we added several sentences to expand the discussion of the impact of native residues on the MS2 CP [lines 326–331]. We also noted that only two cysteines are present in the MS2 CP, thus clarifying our original discussion [lines 331–335].

7. *Method L 284: "This protocol was repeated in full for three total biological replicates". It is not clear to me if it is 1) three totally independent replication, or 2) three consecutive cycle (like phage display library) i.e. use cDNA library of the 1st cycle to generate plasmid library of the 2nd cycle to narrow down toward more mutable variants. (Judging from other part of the manuscript, the first one is probably the case but I am not 100% sure and it would be a good idea to be explicit on this point)*

We added a sentence to clarify that these are three completely independent biological replicates [lines 375–377]

8. *Experimental procedure of a size based selection process is not well described in either main text or supplement. It would be helpful to understand a procedure if a SEC profile*

of VLP mixture is presented and explained what fractions are considered as "assembled" VLP, in comparison with SEC profile of wtVLP.

We added FPLC SEC traces for each replicate as a supplemental figure (Figure S9). These traces are compared to wild-type MS2 with lines indicating where we gated the selection. We also expanded the supplemental methods to include more detail on experimental protocols and computational methods.

9. *Fig 2 and 3: There are two nonsense mutations which have positive fitness score. How is this possible?*

False positives are likely a result of sequencing read errors, as is likely the case for these two stop codons. We added several sentences discussing the impact of sequencing errors on the AFL [lines 236–244]. We also added a high-stringency AFL that eliminated all codons that are one base pair away from the wild-type CP (Figure S6), and trends hold between the two stringencies, leading us to believe that sequencing errors minimally affect the AFL.

Other Minor Points

10. *A more in depth discussion of protein fitness landscapes would be useful and a brief explanation of the modified EMPIRIC cloning method used.*

We expanded the methods section on EMPIRIC cloning [lines 353–355] that describe the method used to generate libraries used in this study. We also added a sentence to more fully describe protein fitness landscapes [lines 48–50].

11. *Most of the first paragraph of the results section is an extension of the introduction. I would suggest removing it from the results section.*

While we agree that the structure is a little nontraditional, we would prefer to keep the first paragraph of the results as is. While the content is important, the information does not quite fit in the flow of the introduction, which is more about VLPs and self-assembly as a whole. We are happy to discuss this further with either reviewers or editors.

12. *L80: A statement of "..... stable towards harsh condition." is a little vague. It would be better to describe more e.g. temperature and pH range etc.*

Wording on lines 91–92 was changed to clarify the conditions that wild-type MS2 can withstand, with two references to support the descriptions:

Hooker, J. M., Kovacs, E. W. & Francis, M. B. Interior Surface Modification of Bacteriophage MS2. *J. Am. Chem. Soc.* **126**, 3718–3719 (2004).

Caldeira, J. C. & Peabody, D. S. Thermal stability of RNA phage virus-like particles displaying foreign peptides. *J. Nanobiotechnology* **9**, 22 (2011).

13. *Figure 1: would benefit from a more in depth caption.*

We expanded the caption in Figure 1 to include more specific information about the selection and mutagenesis strategies.

14. *Figure 2: It is very difficult to distinguish gray color (indicating missing mutations) from blue, particularly pale water color blue. Would be a good idea to use a different color or pattern for missing mutations. e.g. black, or mesh.*

In Figures 2, S2 and S6, we changed the background color to green in order to more clearly mark missing values.

15. *L243-245: It is mentioned that "the two branched residues (I/V) behave more similarly than the two isomers (I/L)". However, Leu also has "branched" side chain like Val and Ile.*

We apologize for this error. Wording was changed on lines 322–323 to more accurately describe our hypothesis regarding the differences between these three amino acids.

16. *"AFL" in L214, L261, and Figure 5 right side legend: Are these a typo of AFS, or does this mean something else?*

We apologize for omitting this definition. AFL is the Apparent Fitness Landscape, while AFS is an individual Apparent Fitness Score. We changed the caption to Figure 5 to clarify that this abbreviation refers to the Apparent Fitness Landscape, and we added a definition to the introduction and abstract [lines 124–125].

17. *Figure 5: There is a typo in the word flexibility (top right).*

We thank the reviewer for noticing this typo, which is fixed.

Reviewer #3 (Remarks to the Author):

The manuscript provides a comprehensive description of mutations on coat protein that impact the assembly of the MS2 capsid. The study design is straightforward and the experiment was executed properly. However the novelty is limited in the current version.

1. *Many studies have presented comprehensive fitness mapping of proteins or viral genomes in the past several years with the advancement of sequencing capacity.*

We agree, and note that we acknowledge these previous studies of enzymes, binding proteins, and viruses in our introduction. This experiment is philosophically different from these other advances in that we are studying the biophysical property of assembly. Thus, we are able to shed light on the sequence-structure relationships that govern macromolecular assembly for the first time in the absence of other phenotypes such as infectivity. We now also show how we can explore the impact of environmental conditions on assembly stability (see response to point 5). With our method, virus-like particles and other protein assemblies can now be directly

engineered for desired structural properties, which is useful for designing vaccines and drug delivery vehicles.

- 2. In this study, the selection is size exclusion chromatography. However the method cannot separate viral capsids from aggregates of similar sizes, which creates background noise.*

To address this and other comments, we added a section to the results to discuss interpretation of the Apparent Fitness Landscape [lines 217–244].

- 3. About 75% of targeted variants were observed in the study, an acceptable coverage, but not ideal.*

We apologize for the confusion. We added a sentence in lines 116–118 to clarify the coverage of the library. While 75% of the targeted variants were observed in the VLP library (following the selection), over 90% of the targeted variants were observed in the plasmid library prior to any selective pressure. The difference between these numbers likely refers to variants that do not form well-formed VLPs and are indicated in red on the Apparent Fitness Landscape.

- 4. Potential explanations for the phenotypes were provided on the basis of biophysical properties of amino acids, consistent with expectations. But the “Apparent fitness score predicts VLP assembly” is an over-statement. It is verification not prediction.*

We apologize for this error. We changed the phrasing from “predicts” to “confirms” to better represent these results [line 183]

- 5. The potential impact on drug delivery is quite distant. If the authors can engineer new capsids with desired properties using the comprehensive fitness map, the novelty and impact of the study will be significantly elevated.*

Virus-like particles, including the MS2 CP, are promising drug delivery vehicles. We performed these analyses in part to enable researchers to rapidly alter the chemical properties of the MS2 CP, improving its viability as a targeted drug delivery vehicle. However, to further clarify the potential of these methods to tune the properties of a VLP, we now describe how we used an acid selection to successfully alter the properties of the MS2 CP [line 264–295]. To do so, we incubated the library in an acidic environment for four hours, then used high-throughput sequencing to identify variants that are sensitive to these conditions. We identified a promising variant, CP[T71H], that is far more sensitive to acidic conditions than the wild-type CP.

REVIEWERS' COMMENTS:

Reviewer #1 (Remarks to the Author):

The revision is satisfactory, so this reviewer suggests the publication of this paper.

Reviewer #3 (Remarks to the Author):

There were a lot inaccurate experiments, interpretations and statements. But the new experiment of screening the mutations at low PH significantly improved the manuscript. Many mistakes were fixed.

Reviewer #4 (Remarks to the Author):

The concerns of Reviewer #2 have been satisfactorily addressed.